# Long-Term Outcomes from a 10-Year Follow-Up of Women Living with a Restrictive Eating Disorder: A Brief Report

**DOI:** 10.3390/nu12082331

**Published:** 2020-08-04

**Authors:** Enza Speranza, Lidia Santarpia, Maurizio Marra, Emilia De Filippo, Olivia Di Vincenzo, Delia Morlino, Fabrizio Pasanisi, Franco Contaldo

**Affiliations:** Internal Medicine and Clinical Nutrition Unit, Department of Clinical Medicine and Surgery, Federico II University Hospital, 80131 Naples, Italy; lidia.santarpia@unina.it (L.S.); marra@unina.it (M.M.); emiliotta@libero.it (E.D.F.); oli.divincenzo@gmail.com (O.D.V.); delia.morlino@unina.it (D.M.); pasanisi@unina.it (F.P.); contaldo@unina.it (F.C.)

**Keywords:** restrictive eating disorders, lifestyle results, bone mineral density

## Abstract

Background: This study aimed to evaluate several socio-demographic and long-term clinical outcomes in a cohort of women living with a restrictive eating disorder. Methods: Patients were asked to fill in a general data collection form aiming to investigate their current conditions and to attend the outpatient unit for a 10-year follow-up clinical and laboratory evaluation. Results: Forty-four patients completed the follow-up general data collection form and 20 agreed to attend the outpatient unit for the 10 year-follow-up evaluation. In total, 52% of patients were single, 55% had achieved a university degree, and 55% had steady employment. After 10 years, there was a clear improvement in biochemical markers, but cholesterol levels were still slightly high. The prevalence of osteopenia in the whole sample was 70% when measured on the lumbar column and 20% on the total body, while osteoporosis was found in 10% of patients and only on the lumbar column. Conclusion: According to the collected data, women with a history of restrictive eating disorders appear to re-adapt well to social life by obtaining the level of their unaffected peers in terms of education and employment.

## 1. Introduction

Anorexia nervosa (AN) is a severe psychiatric disorder mostly observed in young women; it is primarily characterized by the inability to maintain a healthy body weight and also by a distorted body image perception with a fear of becoming fat, despite a low body weight [1]. The overall prevalence has remained stable over the past several decades, while there has been an increased incidence rate in the group of girls aged 15–19 [2]. The disorder, in the long term, is characterized by chronic symptoms with the onset of serious complications [3,4]. The evaluation of relevant aspects of normal life, such as education, employment, marriage, and reproduction, may help to estimate the concerns related to the long-term outcomes of the disorder and to identify possible predictive factors [5]. Studies on the long-term outcomes in patients with AN have demonstrated that in nearly 20% of cases, the disorder becomes chronic, despite many treatment attempts [6,7]. Data on this topic are conflicting, in part due to methodological differences. This study aims to evaluate several socio-demographic (i.e., education, employment, and marriage) and long-term clinical outcomes in a cohort of patients who attended their last outpatient visit for their eating disorder at least 10 years before the study recruitment.

## 2. Materials and Methods

All female outpatients with a diagnosis of restrictive anorexia nervosa (r-AN) according to the former criteria of the Diagnostic and Statistical Manual of Mental Disorders (DSM-IV, in use at the time of the assessment) and confirmed by the recent DSM-V, who visited the Internal Medicine and Clinical Nutrition Unit from January 2000 to December 2005, were recruited for this cohort study.

Patients were asked to fill in a general data collection form (Figure 1) aiming to investigate their current health and socio-demographic factors, eating behavior, and reasons for the follow-up interruption. The protocol of the study was approved by the Local Ethics Committee (prot.N. 37/17), and informed consent was obtained from and signed by all participants.

During their outpatient visits performed at least 10 years before, anthropometric, biochemical, and body composition data were registered. Follow-up data were collected from January to September 2017. Data on death and emigration were obtained from the municipality consultation registry. The follow-up form was administered by an expert dietician, directly at our unit or by telephone/mail. The flow of data collection and the patient participation rates are outlined in Figure 2.

For the 20 patients who attended the outpatient unit for a 10-year follow-up re-evaluation, anthropometric measurements, routine biochemistry, bio-impedance analysis (BIA), indirect calorimetry to evaluate body composition and resting energy expenditure (REE), and dual-energy X-ray absorptiometry (DXA) for bone mineral density (BMD) assessment were performed [8,9].

A specific equation based on body weight and BIA parameters (in particular, the resistance index (RI = height^2^/R, cm^2^/ohm)) was used to estimate the fat-free mass (FFM) [10]. The lumbar and total body T-scores measured by DXA were used for bone density evaluation.

The updated information (Time B) obtained in the group of re-evaluated patients was compared with that collected at least 10 years before (Time A). The statistical analysis was performed by using SPSS version 19.0 (SPSS, Inc., Chicago, IL, USA). The results are expressed as the mean ± standard deviation (SD). One-way analysis of variance (ANOVA) was used for comparison between subgroups, while Tukey’s test was used for pairwise comparisons and the χ^2^ test was used to assess differences in proportions. A *p*-value of <0.05 was considered statistically significant.

## 3. Results

As shown in Figure 2, all 117 female patients living with a restrictive eating disorder who visited the Clinical Nutrition Unit from January 2000 to December 2005 were recruited for the study. Forty-two patients were not reached, due to changing their home address and telephone number; 30 patients refused participation in the study; and one patient had died due to heart failure secondary to severe protein energy malnutrition. Finally, 44 patients (38%; 44/117) completed the follow-up general data collection form, and 20 (45%; 20/44) agreed to attend the outpatient unit for the 10-year follow-up evaluation.

### 3.1. General Data Collection Form

The updated socio-demographic information of the 44 r-AN patients who completed the general data collection form are shown in Table 1. Overall, 23 (52.3%; 23/44) of the patients were single, 24 (55%; 25/44) had achieved a university degree, and 24 (55%; 24/44) had steady employment.

The reported reasons for the interruption of regular visits at the outpatient unit were as follows: 18 patients (40%; 18/44) showed total improvement from the illness, 13 (30%; 13/44) moved to another city, 4 (10%; 4/44) stopped for study/training reasons, and 9 (20%; 9/44) had other reasons. No patients were taking psychotropic drugs; 13 patients (31%; 13/44) were taking drugs for AN-unrelated disorders (two on statins, three on pump inhibitors (PPI), five on hormone replacement therapy, and three on vitamin D supplements). As far as their current general clinical conditions, 22 (50%; 22/44) reported being in good health. During the 10 years prior to the re-evaluation, three patients required hospitalization in a psychiatry ward for their disorder-related complications (no further details available). Meanwhile, seven patients (16%; 7/44) reported irregular menses and 13 (30%; 13/44) regularly practiced mild/moderate physical activity.

### 3.2. Data Comparison

Twenty r-AN patients agreed to attend the outpatient unit for a 10-year follow-up re-evaluation. Table 2 presents a comparison between the anthropometric data, body composition, and REE measured in the 20 r-AN patients at the follow-up visit (Time B) and those measured at least 10 years earlier (Time A).

Weight and body mass index (BMI) tended to increase from Time A to Time B (*p* = 0.002 and 0.001, respectively). At Time B, five patients (25%; 5/20) had a BMI of > 18.5 kg/m^2^. Furthermore, FFM increased (*p* = 0.006), whereas the phase angle (PhA) was not different in the two groups.

The absolute REE was higher at Time B (*p* = 0.023), but this result was not confirmed after the correction for FFM.

### 3.3. Biochemical Abnormalities

The percentages of biochemical abnormalities registered at Time A and at Time B in the sample of 20 r-AN patients are shown in Table 3.

At Time A, serum cholesterol (Chol), amylase, and alkaline phosphatase (ALP) levels were higher than normal in 50%, 80%, and 50% of the patients, respectively; at Time B, only Chol remained slightly higher in 40% of the patients.

### 3.4. Bone Mineral Density

Regarding bone condition, data were available only at the 10-year follow-up. The prevalence of osteopenia in the whole sample was 70% when measured on the lumbar column and 20% on the total body, while osteoporosis was found in 10% of the patients and only on the lumbar column.

## 4. Discussion

Anorexia nervosa is a chronic psychiatric and nutritional disorder that mostly affects adolescents and young women, with high complication and mortality rates [11]. The social impact of the disorder in the short term has been well described [12,13], whereas limited data are available on the long-term clinical and social outcomes [14], with the evaluations mainly limited to death or disability.

An open question of great importance is how women with AN re-adapt to social life after their illness. Only a few studies have evaluated long-term outcomes in AN patients outside of the clinical setting, focusing on the psychological aspects, education, work, relationships, etc. [15].

In our sample, the long-term socio-demographic and lifestyle results of women living with a restrictive eating disorder were investigated using a simple general data form. According to the data reported in the literature in studies with comparable follow-up durations (from 5 to 10 years), as per that in our study, less than half of the patients showed clinical improvement [12].

Our long-term results appear in line with the limited epidemiological data reported in the literature as far as the typical high educational level, which, in turn, reflects the persistence of the typical perfective and apparently firm personality of these patients [12,16]. This finding is also confirmed by the high employment rate compared with that reported for young women in southern Italy (50% vs. 42%) by the Italian National Statistics Institute (ISTAT) [17].

As derived from the questions on drugs taken and the improved body weight, more than half of the patients seem not to have had long-term clinical complications of the disorder. Twenty-eight percent of the interviewed patients were married, which is a lower percentage than in the general population (52%), probably due to the persisting difficulty in socializing and establishing independent relationships with the other sex, even after the active phase of the disorder [18]. According to the collected data, women with a history of a restrictive eating disorder all in all appeared to obtain the level of their unaffected peers in terms of education and employment.

Among the 20 patients undergoing the 10-year follow-up clinical, anthropometric, and laboratory evaluation, in 15/20 (75%) cases, body weight increased with a wide range of variability (from 1 to 20 kg), associated with a trend of increased REE and FFM. Although new interesting results did not emerge for the PhA, we confirmed its role as a predictor of a malnutrition state and as an effective marker of qualitative changes in body composition [9,10,11,12,13,14,15,16,17,18,19].

From the laboratory results obtained in this follow-up study, the biochemical abnormalities (i.e., serum cholesterol, liver transaminases, and pancreatic amylase) found in past evaluations seem to be positively affected by the long-term improvement of body weight.

The finding of near-normality of several functional parameters after the refeeding of malnourished individuals has been already well described by the pioneering Minnesota study by Ancel Keys and co-workers and confirmed by other subsequent studies [20].

Regarding BMD as measured via DXA in the subgroup of 20 r-AN patients, keeping in mind the limited number of patients, the rates of BMD impairment (70% osteopenia and 10% osteoporosis in the whole group) confirm a huge negative effect of restrictive eating disorders on bone health.

Many studies have demonstrated that body weight and, in particular, lean body mass loss strongly impair BMD in AN patients [21,22]. The higher impairment on the lumbar district than the total body seems, at least in part, to be linked to the trabecular structure of vertebral bones and to their higher osteoclast density. In specific conditions, such as in the case of malnutrition due to r-AN, the accelerated bone reabsorption is not adequately balanced out by bone formation [23,24].

Our study has several limitations, starting from the small sample size, explained in part by the type of disorder, which is characterized by poor compliance; this was also confirmed by the high dropout rate at the study time. Despite the small number of participants, we think that the collected data could be informative in dealing with a very-long-term follow-up. Secondly, information was collected by using a general data collection form and not by a standardized assessment test or specific tool. Finally, the lack of initial data on BMD prevented us from evaluating bone changes occurring over time.

Although our sample considered outpatients living with a restrictive eating disorder evaluated at a specialized center, their BMI (37% of patients had a BMI of < 16 kg/m^2^) closely mirrors the complexity of both the disorder and its health risks. In addition, data collection was particularly demanding due to the specific psychological characteristics of r-AN patients, their young age, and, sometimes, their social living context.

## 5. Conclusions

Even though the evaluated sample may appear too small, the homogeneity of the study population and the long time since the last observation (at least 10 years) make our findings reasonably applicable to people living with restrictive eating disorders in southern Europe.

Among the clinical interests of r-AN are the effects of prolonged starvation in adolescence, a crucial period of development and growth associated with important changes in physiological and psychological life; for this reason, preventive strategies, including education programs on healthy eating and psychological and social wellness, are highly recommended.

## Figures and Tables

**Figure 1 nutrients-12-02331-f001:**
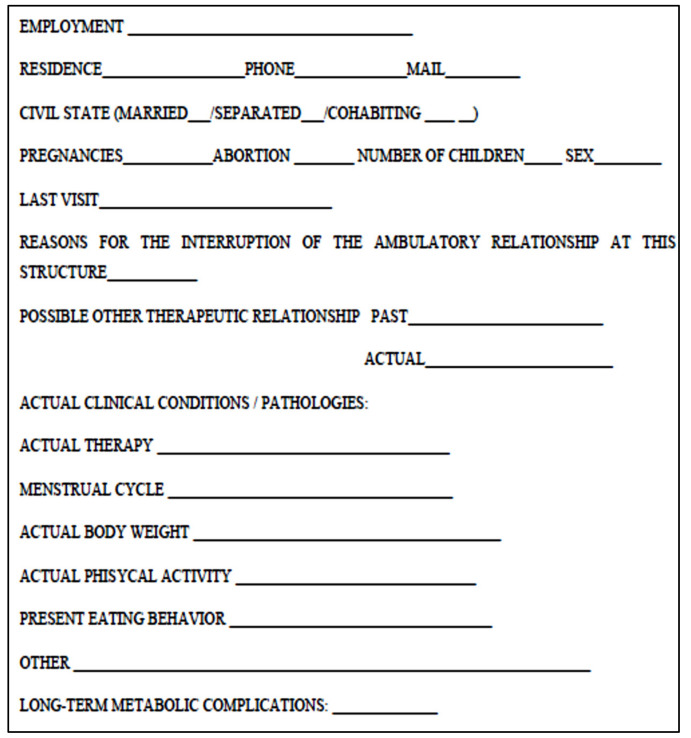
General data collection form.

**Figure 2 nutrients-12-02331-f002:**
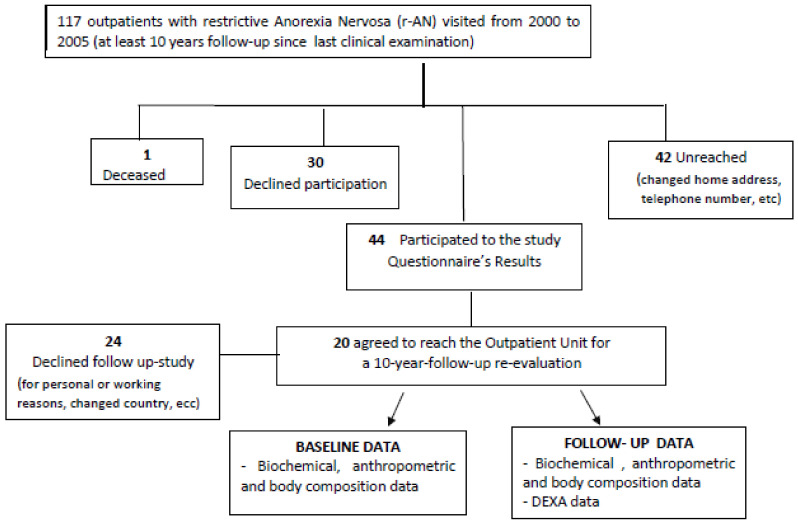
Flow chart describing the different steps of the patients’ involvement in the study.

**Table 1 nutrients-12-02331-t001:** General data collection form results of the 44 r-AN patients at the 10-year follow-up.

Marital Status	*N* (%)	Level of Education	*N* (%)	Employment	*N* (%)
Single	23 (52.5)	Junior high school	9 (20)	Student	9 (20)
Married	12 (27.5)	Diploma high school	11 (25)	Employed	24 (55)
Separated	3 (7.5)	University degree	24 (55)	Housewife	8 (17.5)
Cohabiting	6 (12.5)			Unemployed	3 (7.5)

Data are expressed in numbers and percentages of patients who responded to the general data collection form.

**Table 2 nutrients-12-02331-t002:** Anthropometric and body composition data in the 20 r-AN patients who attended the outpatient unit for the 10-year follow-up re-evaluation—a comparison between current follow-up data (Time B) and data registered at least 10 years earlier (Time A).

Anthropometric and Body Composition Data	Time A	Time B	Difference
Age (years)	20.3 ± 4.8	34.9 ± 5.9	+14.6 ± 2.5 *
Weight (kg)	41.8 ± 5.4	47.2 ± 7.6	+5.4 ± 7.3 *
Stature (cm)	160 ± 6	160 ± 6	-
BMI (kg/m^2^)	16.4 ± 1.6	19.0 ± 2.5	+2.6 ± 2.9 *
FFM (kg)	37.5 ± 4.7	41.1 ± 4.6	+3.7 ± 3.5 *
FM (kg)	4.3 ± 2.1	6.1 ± 3.6	+1.7 ± 3.7
FM (%)	9.9 ± 4.0	12.2 ± 7.2	+2.3 ± 7.3
PhA (degrees)	5.1 ± 1.4	5.2 ± 1.6	+0.14 ± 2.1
REE (kcal/day)	952 ± 168	1130 ± 246	+178 ± 219 *
REE/FFM (kcal/kg)	25.4 ± 3.8	27.5 ± 4.6	+2.1 ± 5.2

Data are expressed as mean ± standard error of the mean (SEM). Comparison between current follow-up data (Time B) and those registered at least 10 years earlier (Time A). Data were analyzed by the paired Student’s *t*-test; * *p* < 0.05. BMI, body mass index; FFM, fat-free mass; FM, fat mass; REE, resting energy expenditure; PhA, phase angle.

**Table 3 nutrients-12-02331-t003:** Rates of biochemical abnormalities in the 20 r-AN patients who attended the outpatient unit for the 10-year follow-up re-evaluation—a comparison between current follow-up data (Time B) and data registered at least 10 years earlier (Time A).

Serum Parameters	Time A (%)	Time B (%)
Hb (<12 g/dL)	20	10
Hct (<35%)	0	0
Lymphocytes (<1200 n/mm^3^)	15	20
Glucose (<60 mg/dL)	5	0
Albumin (<3.6 g/dL)	0	0
Cholesterol (>190 mg/dL)	50	40
AST (>35 UI/L)	5	0
ALT (>35 UI/L)	17	5
ALP (>104 UI/L)	50	5 *
Amylase (<220 UI/L)	80	40 *
PCHE (<5400 UI/L)	30	20
LDH (>400 UI/L)	5	5

Data were analyzed by the chi-square test and are expressed as percentages; * *p* < 0.05. Hb, hemoglobin; Hct, hematocrit; AST, aspartate transaminase; ALP, phosphatase alkaline; PCHE, cholinesterase; LDH, lactate dehydrogenase.

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
