# Peer review of "Long-Term Outcomes from a 10-Year Follow-Up of Women Living with a Restrictive Eating Disorder: A Brief Report"

_nutrients, 2020, doi:10.3390/nu12082331_

Round 1
Reviewer 1 Report
Major comments.
- It is unclear to me how the come up with a ten year follow up, if they have included patients that were at the clinic between 2000 and 2005. Please explain this in a clearer way. Ten year follow up since sickness debut? .. since first visit at clinic?... since last clinical examination? Are they all recovered at follow up? Some seem to still have a low BMI.
- For the follow up group that is split into a group that lost weight and a group that gained weight giving a low n, statistical analysis must be severely underpowered. Thus these “results” should be very cautiously discussed (and not mentioned in the abstract).
- Something weird is going on in line 154-157 “ Authors should discuss the results and how they can be interpreted in perspective of previous studies and of the working hypotheses. The findings and their implications should be discussed in the broadest context possible. Future research directions may also be highlighted.”
- In the discussion I do not think they can draw the conclusion that they have an acceptable quality of life from these data. That requires much more extensive analysis.
- In general, they need to be much more careful with their interpretations since this is a very limited sample set, and they need to be clear on when something is statistically significant (i.e. a difference) or not.
Minor comments:
- In the abstract the sentence “After 10 years, a decrease in the rate of alterations for almost all biochemical abnormalities were evident, except for high serum cholesterol levels whose rate increased in the group who lost weight.” has to be rewritten to be easier to grasp.
- AN is referred to as a disease in several places, but I believe that the correct phrasing is that it is a disorder.
- misspelling of deceased in figure 2
- Is it possible to get cholesterol split into HDL and LDL?
Reviewer 2 Report
Authors should be congratulated on attempting to disseminate very important research and data. Particularly, long-term follow-up data in a sample of individuals with AN which is a challenging feat. Because of this, the publication could be of potential value to the readership of the journal. However, currently there are several serious issues:
In particular, the paper needs careful editing and proof-reading (please note that not all typos and poor sentences structure were pointed out in the suggestions below). The presentation of tables and figures need to be carefully reconsidered. Overall, the paper could do with restructuring to make it more focused, and of value to the readership. As such, it is also recommend this manuscript be re-worked as a brief report, should the journal accept these types of submissions. Further, alongside presenting the preliminary observation, the article should also explain what other data will be presented or analysed in later publications.
Please see our specific comments below.
Abstract
Line 40 – Reconsider use of “suffering.” More appropriate terminology would be, “Patients living with a restrictive eating disorder.”
Introduction
The introduction is brief and does not provide adequate rationale or context for the methods and results that follow.
Line 40 – Reconsider use of “suffering”
Line 41 – it is unclear what is meant by “out of any standardized follow-up” – does this mean patients were not receiving treatment?
Materials and Methods
Insufficient detail is provided in the description of materials and methods and it is unclear what data were collected at the initial assessment.
Line 45 – delete “our” before “Internal Medicine and Clinical Nutrition Unit”
Figure 2 – there are major spelling errors. The figure needs to be reformatted to clearly show which participants (and what data) were captured at baseline, and then follow up. There should also be some mention of why 24 participants did not agree to the 10-year follow up.
Lines 77-79 – additional information is needed on the questionnaire tool used, including, who developed the tool and whether it has been validated or used in previous studies.
Lines 81-87 - briefly describe each method of assessment and provide relevant references.
Lines 88-89 - was eating disorder diagnosis re-assessed at the 10 year follow up? This is highly relevant as it would be interesting to consider the proportion of participants who continue to meet the DMS-5 criteria, even 10 years after initial assessment.
Lines 91-92: Authors should include details of sample size calculation.
Results
Presentation of data should be consistent. Please correct multiple spelling mistakes, and inconsistencies in presentation of abbreviations and acronyms.
Link 105 – Results of “regular pregnancy” are not meaningful here. A more robust questionnaire would have been more valuable to assess if there were participants who wanted to have children but were having troubles/unable to. How long did it take these "regular pregnancies" to actually fall pregnant? Reporting the presence of a "regular pregnancy" is not meaningful if it does not consider other factors in the pregnancy, especially in a population group that is often associated with higher risk for adverse infant and maternal outcomes see: https://onlinelibrary.wiley.com/doi/abs/10.1002/eat.23251
Formatting of tables throughout the results section need to be re-visited and presented more clearly
Table 1: were these sociodemographic data collected from participants at the initial assessment? It is hard to make a meaningful interpretation of this follow-up data if the reader does not know how the participants were tracking when initially receiving treatment for the disorder.
Table 4: Inconsistent use of group a/group b and group 1/group 2.
Table 5: requires significant re-structuring. Further, it does not make sense why the baseline age has been presented alongside data from the follow up.
Within the text, data are inconsistently reported. Please review reporting of n’s, %’s and proportions within the text.
Discussion
Line 187-188: It is unclear how the authors defined “recovery.” Measurement of eating disorder psychopathology, or re-assessment of AN diagnosis at follow up is needed to provide evidence for the statement of “less than half of patients were clinically recovered”
Line 196: A big assumption is made to suggest results imply an “acceptable quality of life.” Authors should have measured quality of life using a validated tool to support this statement.
Line 200: It is unclear what the authors mean by “self-rated health.” A measurement tool for this data is not described within the methodology, and data is not clearly presented in the results.
Line 201-202: 1 sentence is not sufficient to form a paragraph.
Line 208-209: Was there a statistically significant association between PhA and measures of body composition? This has not been presented.
Line 223-226: Interpretation of BMD and DEXA results must be done with caution and clearly align with the fact that this was only measured once. The current wording implies that changes may have occurred over time.
Study limitations
Line 239: Authors mention personality/temperamental traits; however, these data have not been collected, nor reported in within this paper.
There are major limitations in the paper in terms of many measures only being collected and hence reported on at one time point only. Greater characterisation of the sample at baseline is needed, in order to provide context to the changes (if any) that took place over the 10 years following.
Conclusions
Line 249: It is not convincing that the authors have collected data from a population of “stable-severe r-AN.” Some kind of eating disorder psychopathology measure is needed to characterise symptomology of patients at baseline, as well as information around illness duration and previous treatment attempts.
Line 249-250: Authors imply that study participants were receiving treatment for their AN. If this is the case, then treatment should be described within the methods section of the manuscript.
Author Response
Please the attachment
